# Decoding the brain state-dependent relationship between pupil dynamics and resting state fMRI signal fluctuation

**Filip Sobczak[1,2]\*, Patricia Pais-Roldán[1,3], Kengo Takahashi[1,2], Xin Yu[4]\***

[1]Translational Neuroimaging and Neural Control Group, High Field Magnetic Resonance Department, Max Planck Institute for Biological Cybernetics, Tübingen, Germany; [2]Graduate Training Centre of Neuroscience, International Max Planck Research School, University of Tuebingen, Tuebingen, Germany; [3]Institute of Neuroscience and Medicine 4, Medical Imaging Physics, Forschungszentrum Jülich, Jülich, Germany; [4]Athinoula A. Martinos Center for Biomedical Imaging, Massachusetts General Hospital and Harvard Medical School, Charlestown, Massachusetts, United States

**Abstract** Pupil dynamics serve as a physiological indicator of cognitive processes and arousal states of the brain across a diverse range of behavioral experiments. Pupil diameter changes reflect brain state fluctuations driven by neuromodulatory systems. Resting-state fMRI (rs-fMRI) has been used to identify global patterns of neuronal correlation with pupil diameter changes; however, the linkage between distinct brain state-dependent activation patterns of neuromodulatory nuclei with pupil dynamics remains to be explored. Here, we identified four clusters of trials with unique activity patterns related to pupil diameter changes in anesthetized rat brains. Going beyond the typical rs-fMRI correlation analysis with pupil dynamics, we decomposed spatiotemporal patterns of rs-fMRI with principal component analysis (PCA) and characterized the cluster-specific pupil–fMRI relationships by optimizing the PCA component weighting via decoding methods. This work shows that pupil dynamics are tightly coupled with different neuromodulatory centers in different trials, presenting a novel PCA-based decoding method to study the brain state-dependent pupil–fMRI relationship.

**\*For correspondence:**
fsobczak@tue.mpg.de (FS);
XYU9@mgh.harvard.edu (XY)

**Competing interest:** The authors declare that no competing interests exist.

## Introduction

Pupil diameter change reflects the brain state and cognitive processing (*Beatty and Lucero-Wagoner, 2000*; *Eckstein et al., 2017*; *Laeng et al., 2012*; *Wilhelm and Wilhelm, 2003*). It contains information about behavioral variables as diverse as a subject's arousal fluctuation (*McGinley et al., 2015*; *Yoss et al., 1970*; *McCormick et al., 2020*), sensory task performance (*McGinley et al., 2015*; *Hakerem and Sutton, 1966*), movement (*Stringer et al., 2019*; *Salkoff et al., 2020*; *Musall et al., 2019*; *Reimer et al., 2014*), exerted mental effort (*Hess and Polt, 1964*; *Kahneman and Beatty, 1966*; *Alnæs et al., 2014*), expected reward (*O'Doherty et al., 2003*), task-related uncertainty (*Satterthwaite et al., 2007*; *Nassar et al., 2012*; *Richer and Beatty, 1987*), or upcoming decisions (*de Gee et al., 2014*; *Sheng et al., 2020*). This richness of behavioral correlates is partly explained by the fact that multiple neuronal sources drive pupil activity. Pupil diameter changes reflect spontaneous neural activity across the cortex (*Stringer et al., 2019*; *Salkoff et al., 2020*; *Musall et al., 2019*; *Yellin et al., 2015*; *Pais-Roldán et al., 2020*) and in major subcortical areas (*Stringer et al., 2019*; *Joshi et al., 2016*; *Wang et al., 2012*; *Schneider et al., 2016*; *Ranson and Magoun, 1933*). Both sympathetic and parasympathetic systems innervate muscles controlling pupil dilation and constriction (*Bonvallet and*

*Zbrozyna, 1963*; *McDougal and Gamlin, 2015*; *Yüzgeç et al., 2018*), and the activity of subcortical nuclei mediating neuromodulation has been tightly coupled with pupillary movements (*Pais-Roldán et al., 2020*; *Joshi et al., 2016*; *Reimer et al., 2016*; *Rajkowski, 1993*; *de Gee et al., 2017*; *Murphy et al., 2014*; *Breton-Provencher and Sur, 2019*). In particular, rapid and sustained pupil size changes are associated with cortical noradrenergic and cholinergic projections, respectively (*Reimer et al., 2016*), and direct recordings of the noradrenergic locus coeruleus demonstrate neuronal activity highly correlated with pupil dynamics (*Joshi et al., 2016*; *Rajkowski, 1993*; *Breton-Provencher and Sur, 2019*; *Aston-Jones and Cohen, 2005*). Also, pupil diameter changes are regulated through dopaminergic neuromodulation under drug administration (*Shannon et al., 1976*) and in reward-related tasks (*O'Doherty et al., 2003*; *de Gee et al., 2017*). Studies also show that pupil dilation and constriction can be controlled by serotonergic agonists and antagonists, respectively (*Vitiello et al., 1997*; *Schmid et al., 2015*). These studies have revealed the highly complex relationship between pupil dynamics and brain state fluctuations (*McCormick et al., 2020*; *Reimer et al., 2014*; *Yüzgeç et al., 2018*; *Lowenstein et al., 1963*).

Resting-state fMRI (rs-fMRI) studies have uncovered global pupil–fMRI correlation patterns in human brains as well as revealed that the pupil dynamics–fMRI relationship changed under different lighting conditions or when subjects engaged in mental imagery (*Yellin et al., 2015*; *Schneider et al., 2016*). The dynamic functional connectivity changes detected by fMRI, possibly modulated by the interplay of cholinergic and noradrenergic systems (*Shine, 2019*), are also reflected in pupil dynamics both at rest (*Shine et al., 2016*) and in task conditions (*Mäki-Marttunen, 2020*). Furthermore, rs-fMRI has been used to display a differential correlation pattern with brainstem noradrenergic nuclei, e.g., A5 cell group, depending on the cortical cross-frequency coupling state in the animal model (*Pais-Roldán et al., 2020*). Although rs-fMRI enables brain-wide pupil–fMRI correlation analysis across different states, the linkage of brain state-dependent pupil dynamics with distinct activation patterns of neuromodulatory nuclei remains to be thoroughly investigated beyond the conventional analysis methods.

Here, we aimed to differentiate brain states with varied coupling patterns of pupil dynamics with the subcortical activity of major neuromodulatory nuclei in an anesthetized rat model. First, we demonstrated that the pupil–fMRI relationship is not uniform across different scanning trials and employed a clustering procedure to identify distinct pupil–fMRI spatial correlation patterns from a cohort of datasets. Next, we modeled the relationship of the two modalities for each cluster using principal component analysis (PCA)-based decoding methods (gated recurrent unit [GRU] [*Cho, 2014*] neural networks and linear regression) and characterized unique subcortical activation patterns coupled with specific pupil dynamic features. This work demonstrates the effectiveness of PCA-based decoding

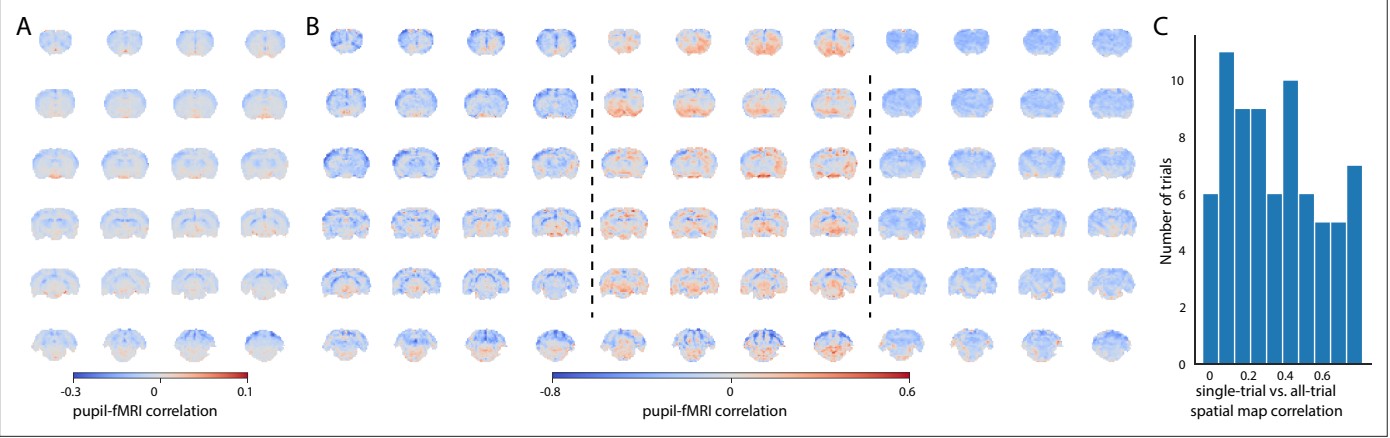

**Figure 1.** Variability of the pupil–fMRI linkage. (**A**) Pupil–fMRI correlation map created by correlating the two modalities' concatenated signals from all trials. (**B**) Selected individual-trial correlations maps. (**C**) Histogram of spatial correlations between the all-trial correlation map and individual-trial maps. High variability of similarities between the maps shows that the pupil–fMRI relationship is not stationary and changes across trials.

The online version of this article includes the following source data for figure 1:

**Source data 1.** The mean correlation map (A), all individual correlation maps (B), and the spatial correlation values (C) are available in the source data file.

to dissect the time-varied pupil–fMRI relationship corresponding to different forms of brain state-dependent neuromodulation.

## Results
### Identification of brain states with distinct pupil dynamics correlation patterns

To investigate brain state-dependent pupil dynamics, we acquired whole-brain rs-fMRI with real-time pupillometry in anesthetized rats (n = 10) as previously reported (*Pais-Roldán et al., 2020*). Initially, the pupil dilation and fMRI time series from all 15 min trials (n = 74) were concatenated. A voxel-wise correlation map of the concatenated pupil signals with fMRI time courses showed a global negative correlation (*Figure 1A*). However, the generated map was not representative of all trials, which was revealed by creating correlation maps for individual trials (*Figure 1B*). These maps demonstrated high variability of pupil–fMRI correlations, which is presented by the histogram distribution of spatial correlation values between individual-trial spatial maps and the concatenated all-trial map (*Figure 1C*).

Next, we clustered all trials into different groups based on pupil–fMRI correlation maps (*Figure 2A*). To facilitate the clustering analysis, we reduced the dimensionality of the spatial correlation maps using the uniform manifold approximation and projection (UMAP) method (*McInnes et al., 2020*) and decreased the number of features used for clustering from the number of voxels (n = 20,804) to 72 for each map. Three to seven clusters were identified with Gaussian mixture modeling and examined using silhouette analysis (*McLachlan and Basford, 1988*; *Rousseeuw, 1987*). Here, we focused on the four-cluster categorization since this division yielded the highest mean silhouette scores (*Figure 2B*) across 100 random UMAP and GMM initializations. For each trial, we selected its most common cluster membership across the 100 repetitions and used it in the following analysis. The clustering results exhibited a very high degree of reproducibility as seen in the plots displaying the reproducibility of cluster labels and mean cluster correlation maps (*Figure 2—figure supplement 1*). The clusters had the following trial counts: $n_1 = 8$; $n_2 = 30$; $n_3 = 24$; $n_4 = 12$. The mean power spectral density (PSD) estimates of pupil dynamics based on the cluster division were plotted in *Figure 2C*. PSD of cluster one showed a distinct peak at 0.018 Hz as well as the lowest baseline pupil diameter values. In contrast, cluster 4 had the highest mean baseline diameter and a peak at 0.011 Hz. Clusters 2 and 3 showed peaks of oscillatory power at less than 0.01 Hz. The ultra-slow oscillation is typical for spontaneous pupil fluctuations (*McLaren et al., 1992*). All PSDs and example pupil signals from each cluster are shown in *Figure 2—figure supplement 2*. We recreated pupil–fMRI correlation maps based on the four clusters (*Figure 2D*). Three clusters (1, 2, and 4) showed negative correlations across large parts of the brain, with the correlation strength differing across clusters. In contrast, cluster 3 displayed a very low mean correlation with positive coefficients spreading across the entire brain. It is also noteworthy that cluster 1 showed a high positive correlation in the periaqueductal gray and ventral midbrain regions. The distinct qualities of identified clusters supported the usage of data-driven clustering for identifying brain state-dependent pupil dynamics.

Lastly, we performed a series of analyses to investigate cluster reproducibility beyond the initial random initializations. First, to compensate for the possible lag between pupil and fMRI, we convolved pupil signals with hemodynamic response function (HRF) kernels with different peak times (*Yellin et al., 2015*; *Pais-Roldán et al., 2020*; *Figure 2—figure supplement 3A*). We regenerated the correlation maps and repeated the clustering procedure 100 times for each kernel. The high cluster membership and correlation map reproducibility across a range of HRF peak times (*Figure 2—figure supplement 3B,C*) justify the use of non-convolved signals and emphasize the impact of slow fluctuations on the correlation results. Similarly to *Allen et al., 2014*, we performed 100 half-split reproducibility analyses and showed that to a large degree the cluster memberships are preserved when using half of the trials (*Figure 2—figure supplement 4*). The match might be imperfect, e.g., due to smaller numbers of a particular cluster's samples in a half-split interacting with UMAP dimensionality reduction parameters. Next, using spatial surrogate maps with spatial autocorrelation and value distribution matching that of real correlation maps (*Burt et al., 2020*) (see Materials and methods), we verified that the spatial location of correlation values and not the mean values or spatial autocorrelation properties were driving the clustering (*Figure 2—figure supplement 5*). Finally, we showed that when splitting the trials into shorter runs, clustering the data into n = 4 clusters should be selected based on the silhouette score

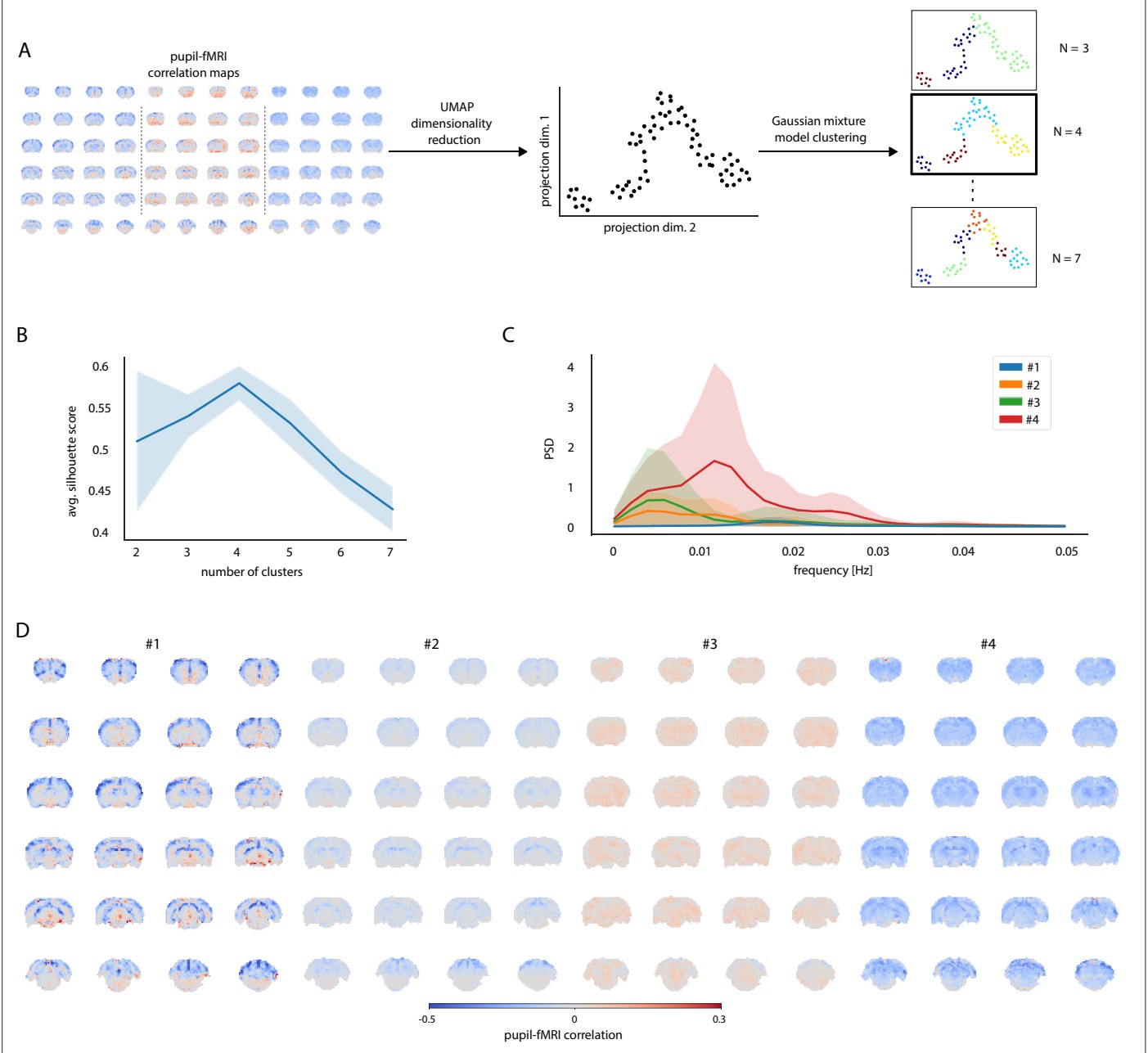

**Figure 2.** Clustering of trials with distinct pupil–fMRI correlation patterns. (**A**) Schematic of the clustering procedure. UMAP is used to reduce the dimensionality of all individual-trial correlation maps to 72 dimensions. A 2D UMAP-projection of the real data is shown. Each dot represents a single trial. The trials are clustered using Gaussian mixture model clustering. Different numbers of clusters are evaluated. (**B**) The final number of clusters is selected based on silhouette analysis. The highest average silhouette score is obtained with k = 4 clusters. Shaded area shows standard deviations. (**C**) Pupil power spectral density estimates (PSD) of each of the four clusters. Signals were downsampled to match the fMRI sampling rate. Shaded areas show standard deviations. (**D**) Cluster-specific correlation maps based on concatenated signals belonging to the respective groups.

The online version of this article includes the following source data and figure supplement(s) for figure 2:

**Source data 1.** Cluster trial labels, individual silhouette scores (B), mean cluster PSDs (C), and cluster-specific correlation maps (D) are available in the source data file.

**Figure supplement 1.** Cluster reproducibility across 100 repetitions with random UMAP and GMM initializations.

**Figure supplement 1—source data 1.** The label match ratios (A) and map similarity values (B) are available in the source data file.

**Figure supplement 2.** Cluster-specific pupil fluctuation features.

*Figure 2 continued on next page*

*Figure 2 continued*

**Figure supplement 2—source data 1.** All PSDs (A) are available in the source data file.

**Figure supplement 3.** Clustering reproducibility across 100 clustering repetitions based on HRF-convolved pupil signals.

**Figure supplement 3—source data 1.** The HRF kernels (A), cluster membership label match ratios (B), and map similarity values (C) are available in the source data file.

**Figure supplement 4.** Cluster reproducibility across 100 repetitions of split-halves clustering.

**Figure supplement 4—source data 1.** The label match ratios (A) and map similarity values (B) are available in the source data file.

**Figure supplement 5.** Cluster reproducibility across 100 sets of artificially generated surrogates with values and spatial autocorrelations matching those of real maps.

**Figure supplement 5—source data 1.** Ten example surrogate sets (i.e. 740 maps total) (A), label match ratios (B), and map similarity values (C) are available in the source data file.

**Figure supplement 6.** Mean silhouette scores based on 100 clustering repetitions performed on shorter trials.

**Figure supplement 6—source data 1.** Individual silhouette scores are available in the source data file.

---

criterion up to a 300 s trial length (*Figure 2—figure supplement 6*). The conducted analyses further justified the selection of n = 4 clusters and verified the reproducibility of the UMAP and GMM clustering procedure.

## Decoding-based investigation of the relationship between whole-brain rs-fMRI and pupil dynamics

To characterize the pupil–fMRI relationship beyond the conventional correlation analysis, we implemented data-driven decoding models to couple the dynamics of the two modalities. First, we validated the approach in a setting involving all trials. We then employed it to investigate the pupil–fMRI coupling in the previously identified clusters. First, we performed principal component analysis (PCA) to extract spatiotemporal features of whole-brain rs-fMRI signals (n = 300) and trained either linear regression (LR) or a gated recurrent unit (GRU) neural network to predict pupil dynamics based on rs-fMRI PCA time courses (*Figure 3A*). Furthermore, we compared the LR and GRU prediction models with a correlation-template-based pupil dynamics estimation used in previous studies (*Pais-Roldán et al., 2020*; *Chang et al., 2016*). All methods were trained on randomly chosen 64 trials using cross-validation and then were tested on additional 10 unseen trials from the same animals. The PCA model was fit with the 64 training trials only. As the correlation-template-based predictions were bounded to the $<-1; 1>$ range, Pearson's correlation coefficient was used to evaluate the decoding of all methods. We optimized the hyperparameters of GRUs and linear regression variants using Bayesian optimization and fourfold cross-validation (hyperparameter values are listed in Materials and methods). Both linear regression and GRU outperformed the correlation-template approach on both training ($CC_{base} = 0.37 \pm 0.27$ s.d., $CC_{LR} = 0.45 \pm 0.26$ s.d., $CC_{GRU} = 0.46 \pm 0.25$ s.d., $p_{LR} = 4.3*10^{-6}$, $p_{GRU} = 2.4 \times 10^{-6}$) and test sets ($CC_{base} = 0.25 \pm 0.17$ s.d., $CC_{LR} = 0.44 \pm 0.24$ s.d., $CC_{GRU} = 0.45 \pm 0.27$ s.d., $p_{LR} = 0.003$, $p_{GRU} = 0.01$) (*Figure 3B*). Notably, the test set prediction scores do not reflect generalization across different rats as the training and test data could belong to the same animals. We repeated the linear regression prediction procedure (including the PCA step) on 100 other random train-test trial splits and validated that the obtained scores are representative of the distribution (*Figure 3—figure supplement 1A*). We also verified the number of rs-fMRI PCA components by testing varied component counts, showing that the highest prediction scores were achieved with 300 components (*Figure 3—figure supplement 1B,C*). In addition, when varying the temporal shift between pupil dynamics and rs-fMRI signals, we obtained the highest prediction scores with zero shift between the input and output signals (*Figure 3—figure supplement 1B*). Similarly, the highest prediction scores were obtained based on pupil signals convolved with an HRF kernel with a peak at 0 s (*Figure 3—figure supplement 1C*). Interestingly, the component which explained the most pupillary variance (explained var. = 7.03%) and had the highest linear regression weight, explained only 0.8 % of the fMRI variance (*Figure 3—figure supplement 2*). Furthermore, the component that explained the most fMRI variance (explained var. = 22.01%) was weakly coupled with the pupil fluctuation (explained var. = 0.51%). Thus, this prediction-based PCA component weighting scheme enabled the dissection of unique brain activity features for the modeling of the pupil–fMRI relationship. It should also be noted that GRU and linear

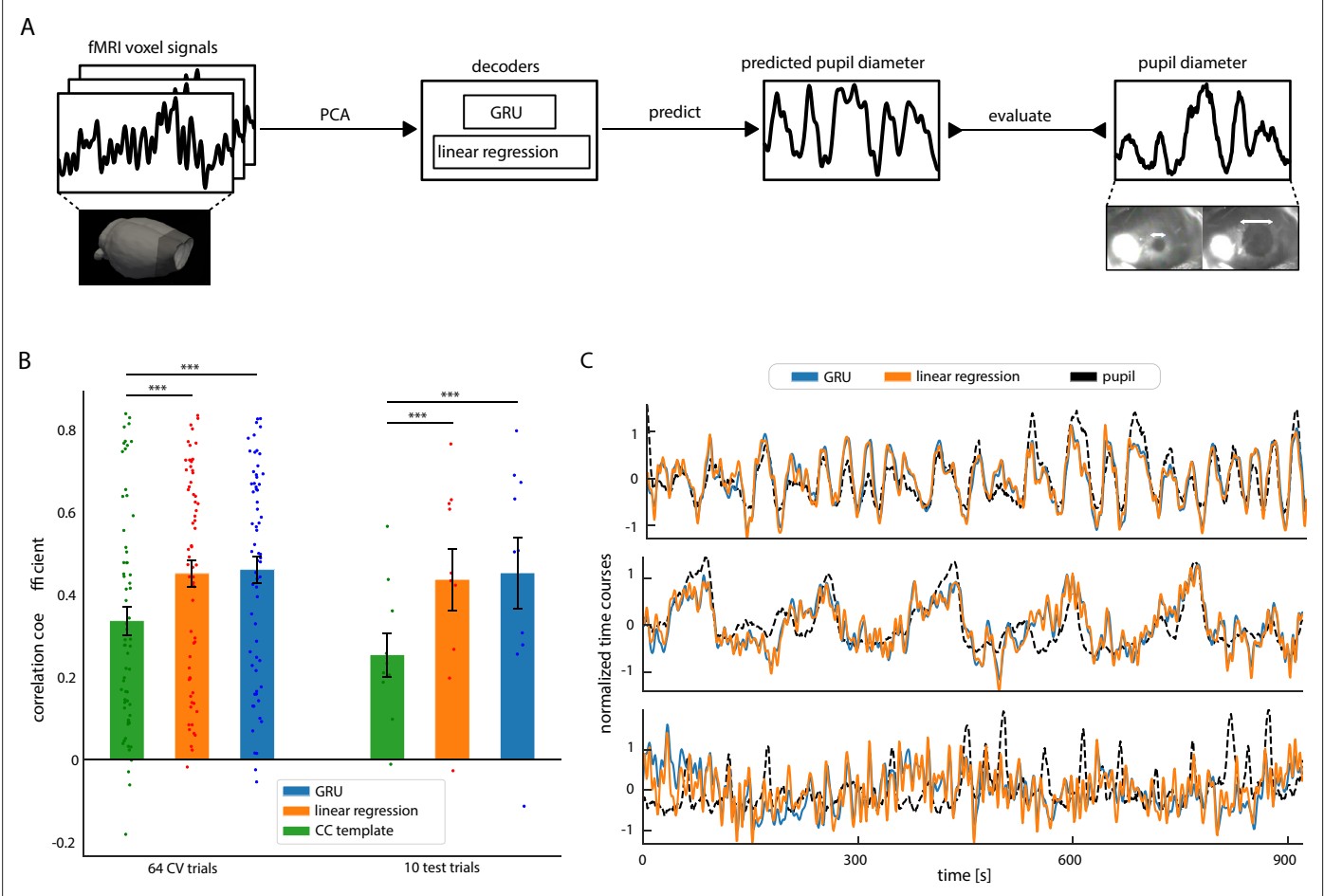

**Figure 3.** Decoding pupil dynamics based on fMRI signals. (**A**) Schematic of the decoding procedure. PCA was applied to fMRI data. The PCA time courses were fed into either linear regression or GRU decoders, which generated pupil signal predictions. The prediction quality was evaluated by comparing the generated signals with real pupil fluctuations using Pearson's correlation coefficients. (**B**) Comparison of the three methods' pupil dynamics predictions. Linear regression and GRU performed better than the correlation-based baseline method on both the cross-validation splits ($CC_{base} = 0.37 \pm 0.27$ s.d., $CC_{LR} = 0.45 \pm 0.26$ s.d., $CC_{GRU} = 0.46 \pm 0.25$ s.d., $p_{LR} = 4.3*10^{-6}$, $p_{GRU} = 2.4 \times 10^{-6}$) and on test data ($CC_{base} = 0.25 \pm 0.17$ s.d., $CC_{LR} = 0.44 \pm 0.24$ s.d., $CC_{GRU} = 0.45 \pm 0.27$ s.d., $p_{LR} = 0.003$, $p_{GRU} = 0.01$). Scattered points show individual prediction scores. (**C**) Linear regression and GRU predictions of three selected trials ($CC_{GRU-top} = 0.79$, $CC_{LR-top} = 0.77$, $CC_{GRU-middle} = 0.75$, $CC_{LR-middle} = 0.73$, $CC_{GRU-bottom} = 0.02$, $CC_{LR-bottom} = 0.06$). Qualitatively, linear regression and GRU predictions were very similar.

The online version of this article includes the following source data and figure supplement(s) for figure 3:

**Source data 1.** Prediction scores (B) and all predicted time courses (C) are available in the source data file.

**Figure supplement 1.** Prediction score dependence on train-test split trial selection, number of PCA components, and temporal shifts between pupil and fMRI signals.

**Figure supplement 1—source data 1.** Individual prediction scores across all trial mixes (A) and the mean prediction scores (BC) are available in the source data file.

**Figure supplement 2.** PCA decoupling of pupil-related fMRI activity from other signal sources.

**Figure supplement 2—source data 1.** The variance explained values (AB) and linear regression weights (C) are available in the source data file.

**Figure supplement 3.** Similarity of GRU and linear regression prediction maps.

**Figure supplement 3—source data 1.** The maps are available in the source data file.

regression methods obtained comparable scores and both methods showed similar prediction performance (*Figure 3C*). *Figure 3—figure supplement 3* shows prediction maps created by integrating PCA components using linear regression weights or averaged GRU gradients (details in Materials and methods). The resemblance of the two maps suggests that despite GRU's potential for encoding

complex and non-linear functions, a linear regression-based rs-fMRI mapping scheme was sufficient for predicting pupil dynamics.

The map generated by combining PCA components with the linear regression decoder enabled the identification of brain nuclei which were not highlighted in the correlation map shown in *Figure 1A*. *Figure 4* shows an overview of the PCA-based fMRI prediction map overlaid on the brain atlas, revealing pupil-related activation patterns at key neuromodulatory nuclei of the ascending reticular activating system (ARAS) – the dopaminergic ventral tegmental area, substantia nigra and supramammillary nucleus, the serotonergic raphe and B9 cells, the histaminergic tuberomammillary nucleus, the cholinergic laterodorsal tegmental and pontine nuclei, the glutamatergic parabrachial nuclei, and the area containing the noradrenergic locus coeruleus. Positive weights were also located in subcortical regions involved in autonomous regulation – the lateral and preoptic hypothalamus and the periaqueductal gray. In addition, the subcortical basal forebrain nuclei (the horizontal limb of the diagonal band, nucleus accumbens, and olfactory tubercle) and the septal area were positively coupled with pupil dynamics. Lastly, regions of the hippocampal formation – the hippocampus, entorhinal cortex, and subiculum, as well as cingulate, retrosplenial, and visual cortices displayed positive weighting. It should be noted that the thalamus and the hippocampus displayed both positive and negative weights. Negative coupling was also found in the cerebellum and most somatosensory cortical regions. The voxel-wise statistical significance (p<0.01) was validated using randomization tests and corrected for multiple comparisons with false discovery rate correction (details in Materials and methods). The identification of pupil-related information in brain regions closely tied to neuromodulatory activity and to autonomous and brain state regulation (*Duyn et al., 2020*; *Benarroch, 2018*; *Dampney, 2016*; *Silvani et al., 2015*; *Kuwaki and Zhang, 2010*; *Grimaldi et al., 2014*; *van den Brink et al., 2019*) highlights the advantage of using PCA decomposition combined with prediction-based decoding methods instead of conventional correlation analysis to identify pupil-related subcortical activation patterns.

## Characterization of brain state-dependent PCA-based pupil–fMRI prediction maps

To differentiate brain state-dependent subcortical activation patterns related to different pupil dynamics, we retrained the linear regression model based on the four different clusters shown in *Figure 2D* and created PCA-based fMRI prediction maps for each cluster (*Figure 5*).

Each PCA-based prediction map portrayed a cluster-specific spatial pattern (*Figure 5B*). Cluster 1 was characterized by strong positive weights in the dopaminergic substantia nigra and ventral tegmental area as well as in their efferent projections in the striatum (nucleus accumbens and caudate-putamen) (*Beckstead et al., 1979*). Positive coupling was also displayed in the periaqueductal gray and brainstem laterodorsal tegmental and parabrachial nuclei as well as in the superior colliculus. Cluster 2 had the strongest positive weights in hypothalamic regions, lateral in particular, but also in brainstem arousal-regulating areas containing the locus coeruleus, laterodorsal tegmental, and parabrachial nuclei. High positive values were also found in the septal area and the olfactory tubercle. In cluster 3, the highest values were visible in preoptic and other hypothalamic areas, as well as in stria terminalis carrying primarily afferent hypothalamic fibers (*De Olmos and Ingram, 1972*), caudate-putamen, and globus pallidus. As in cluster 2, the region containing the locus coeruleus, laterodorsal tegmental, and parabrachial nuclei showed positive linkage with pupil dynamics. Contrastingly, in cluster 4, caudal raphe was the neuromodulatory region showing the strongest positive weights and the anterior parts of the brainstem displayed negative weighting. Characteristic to cluster 4 were high weights in the hippocampus and the subiculum forming the hippocampal formation, as well as in thalamic and amygdaloid areas. In all clusters, negative weights were detected across somatosensory cortices, the cerebellum, and posterior parts of the thalamus, as well as positive weights in hypothalamic and anterior thalamic nuclei and in the area containing the tuberomammillary nucleus. The subiculum and parts of the hippocampus were also positive in all clusters; however, the entorhinal cortex, also belonging to the hippocampal formation, was positive only in clusters 1–3. The same three clusters showed major positive weights in the neuromodulatory brainstem regions, substantia nigra, and ventral tegmental area. Clusters 2–4 displayed strong weights in the supramammillary nucleus, retrosplenial cortex, and the cingulate cortex, which has been coupled with both noradrenergic modulation (*Aston-Jones and Cohen, 2005*) and pupil dynamics (*Pais-Roldán et al., 2020*; *Joshi et al., 2016*). Enlarged cluster-specific maps are displayed in *Figure 5—figure supplements*

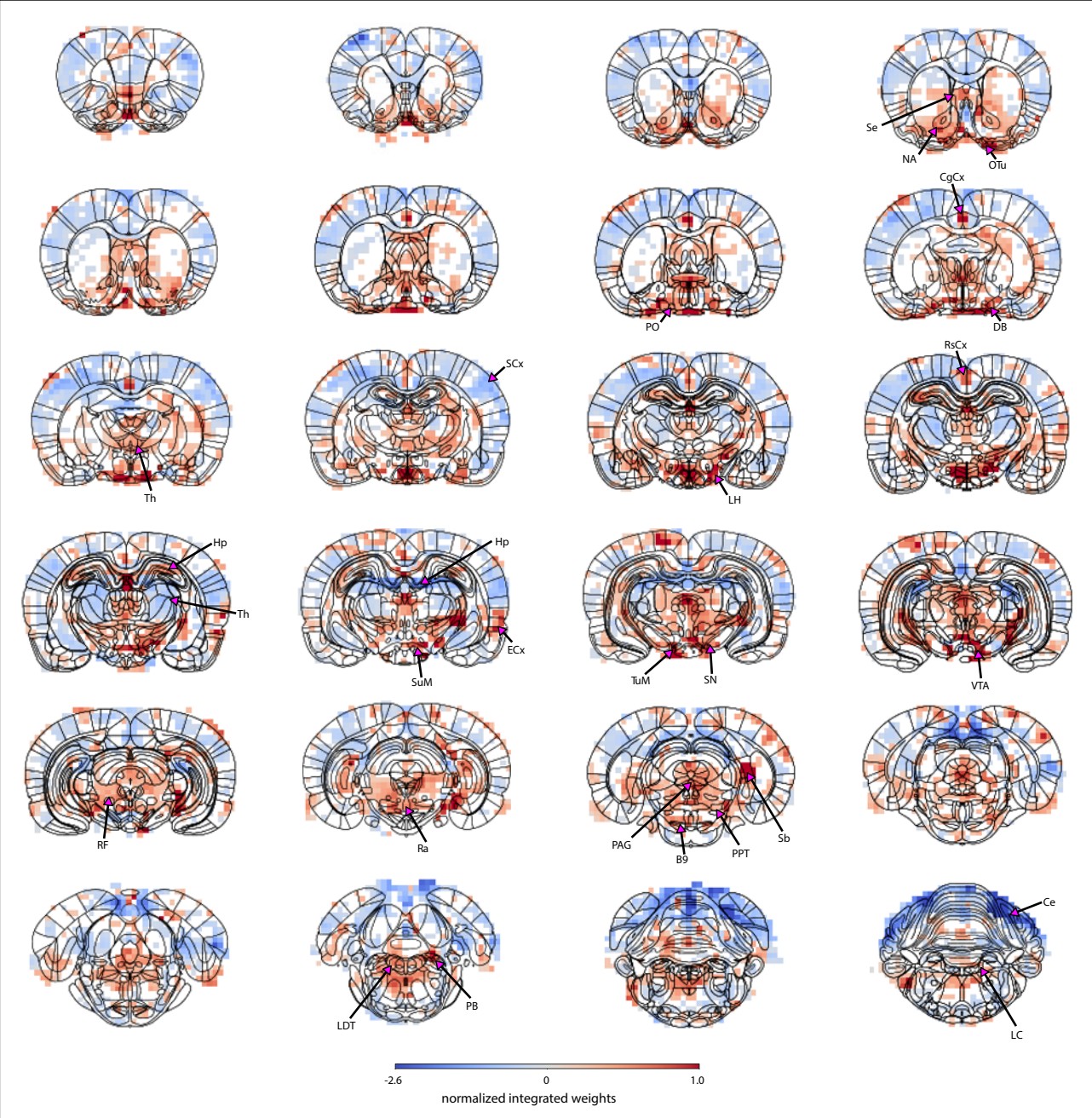

**Figure 4.** Localization of pupil dynamics-related information content across the brain. The spatial map highlights regions from which pupil-related information was decoded. It was created by integrating PCA spatial maps with weights of the trained linear regression model. The map displays positive weights in all neuromodulatory regions of the ascending reticular activating system as well as in other regions involved in autonomous regulation – the lateral and preoptic hypothalamus and the periaqueductal gray. The subcortical basal forebrain nuclei (the horizontal limb of the diagonal band, nucleus accumbens, and olfactory tubercle) and the septal area were also positively coupled to pupil dynamics. Finally, regions of the hippocampal formation – the hippocampus, entorhinal cortex and subiculum, as well as cingulate, retrosplenial and visual cortices showed positive weights. The thalamus and the hippocampus had both positive and negative weights. Strong negative weighting was found in the cerebellum and most somatosensory cortical regions. Masked regions (white) did not pass the false discovery rate corrected significance threshold (p=0.01). Abbreviations: B9 – B9 serotonergic cells, Ce – cerebellum, CgCx – cingulate cortex, DB – horizontal limb of the diagonal band, ECx – entorhinal cortex, Hp – hippocampus, LC – locus coeruleus, LDT – laterodorsal tegmental nuclei, LH – lateral hypothalamus, NA – nucleus accumbens, OTu – olfactory tubercle, PAG – periaqueductal gray, PB – parabrachial nuclei, PO – preoptic nuclei, PPT – pedunculopontine tegmental nuclei, Ra – raphe, RF – reticular formation, RsCx – retrosplenial cortex, Sb – subiculum, SCx – somatosensory cortex, Se – septal nuclei, SN – substantia nigra, SuM – supramammillary nucleus, Th – thalamus, TuM – tuberomammillary nucleus, VTA – ventral tegmental area.

The online version of this article includes the following source data for figure 4:

*Figure 4 continued on next page*

*Figure 4 continued*
**Source data 1.** The masked map is available in the source data file.

*1–4*. Here, we demonstrated the effectiveness of combining the PCA-based approach with clustering methods to reveal brain state-specific subcortical activity patterns related to pupil diameter changes.

## Discussion

Previous studies analyzed the relationship of fMRI and pupil dynamics either by directly correlating pupil size changes with the fMRI signal fluctuation (*Yellin et al., 2015*; *Pais-Roldán et al., 2020*; *Schneider et al., 2016*) or by applying a general linear model to produce voxel-wise activation maps (*Alnæs et al., 2014*; *Murphy et al., 2014*; *Clewett et al., 2018*). Here, we performed PCA-based dimensionality reduction to decouple spatiotemporal features of fMRI signals (*Mwangi et al., 2014*) and implemented prediction methods to decode pupil dynamics based on the optimized PCA component weighting (*Figure 3*).

Two advantages can be highlighted in the present pupil–fMRI dynamic mapping scheme. First, conventional correlation analysis relies on the temporal features of fMRI time courses from individual voxels or regions of interest. Hence, it could not decouple the superimposed effects of multiple signal sources (*Carbonell et al., 2011*; *Tong et al., 2019*) or characterize the state-dependent dynamic subcortical correlation patterns. On the other hand, the PCA decomposition scheme solved these issues by decoupling multiple components of rs-fMRI signals with unique spatiotemporal patterns carrying pupil-related information. Second, the data-driven training of prediction methods optimized the weighting of individual rs-fMRI PCA components. Using the optimized neural network (GRU) or linear regression (LR)-based decoding models, we created prediction maps linking pupil dynamics with fMRI signal fluctuation of specific subcortical nuclei (*Figure 4*, *Figure 3—figure supplement 3*). Also, the decoding models showed much better pupil dynamics prediction than the correlation-template-based approach reported previously (*Pais-Roldán et al., 2020*; *Chang et al., 2016*). Meanwhile, it should be noted that both LR and GRU models generated qualitatively similar prediction maps, highlighting the pupil-related rs-fMRI signal fluctuation from the same subcortical brain regions (*Figure 3C*, *Figure 3—figure supplement 3*). Unlike our previous single-vessel fMRI prediction study (*Sobczak et al., 2021a*), the GRU-based neural network prediction scheme may require much bigger training datasets to outperform linear regression modeling (*Schulz et al., 2020*). Another plausible explanation is that the pupil dynamics were predominantly and linearly driven by only a few rs-fMRI PCA components (*Figure 3—figure supplement 2*), presenting brain activation patterns related to arousal fluctuation and autonomous regulation (*Duyn et al., 2020*; *Özbay, 2019*).

The PCA-based prediction modeling provides a novel scheme to decipher subcortical spatial patterns of fMRI signal fluctuation related to brain state-dependent pupil dynamics. Most notably, neuromodulatory nuclei of ARAS and other subcortical nuclei involved in brain state modulation, as well as autonomous regulation were identified in the PCA-prediction map created from all trials. The highlighted hypothalamus, basal forebrain, and neuromodulatory brainstem nuclei are responsible for both global brain state modulation and autonomous cardiovascular, respiratory, and baroreflex control (*Duyn et al., 2020*; *Benarroch, 2018*; *Dampney, 2016*; *Silvani et al., 2015*; *Kuwaki and Zhang, 2010*; *Grimaldi et al., 2014*; *van den Brink et al., 2019*). Consequently, the source of pupil-related information found across the cortex was probably modulated through global subcortical projections rather than a more direct causal interaction with pupil size changes (*Reimer et al., 2016*; *Lecrux and Hamel, 2016*). Noradrenergic neurons of the locus coeruleus are the hypothesized drivers of pupil dilation (*Joshi et al., 2016*; *Aston-Jones and Cohen, 2005*), and both the area containing the locus coeruleus and many of its input regions (*Breton-Provencher and Sur, 2019*) were highlighted in the PCA map. However, the observed activation of the hypothalamus and other neuromodulatory nuclei suggests that, in the anesthetized state, pupil diameter fluctuation reflects a complex interaction of subcortical homeostatic and brain state-modulating centers.

Also, we have shown that these subcortical interactions and the neural correlates of pupil dynamics are not stationary but change across trials in a brain state-dependent manner. Based on the correlation patterns, we identified four clusters of trials with distinct pupil–fMRI coupling. The clusters displayed a high degree of reproducibility when repeating the clustering procedure with all trials; however, the

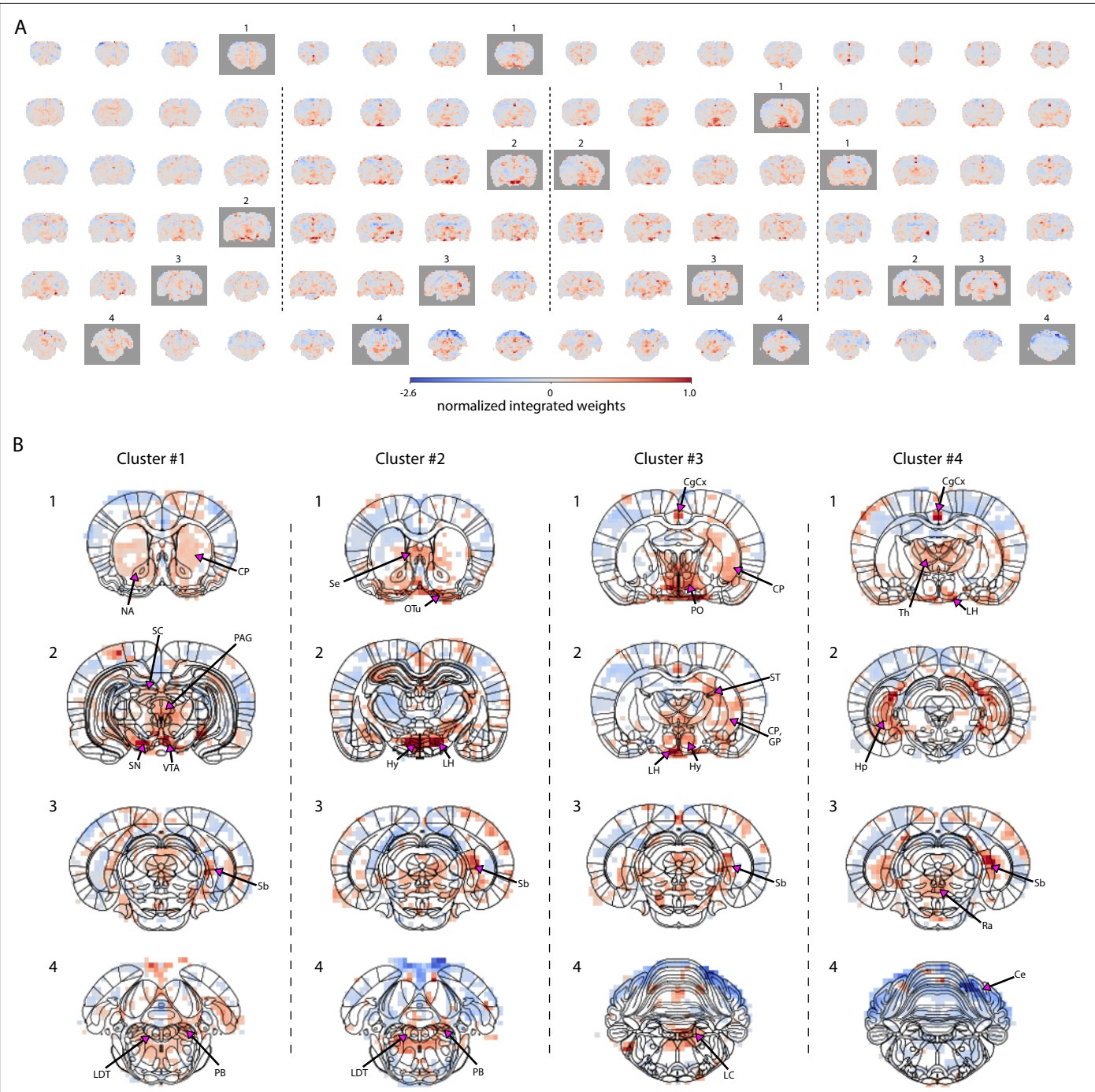

**Figure 5.** Characterization of brain state-specific pupil–fMRI relationships. (**A**) Pupil information content maps generated by integrating PCA spatial maps with weights of linear regression models trained on cluster-specific trials. In all clusters, negative weights were found in the somatosensory cortex, the cerebellum, and posterior parts of the thalamus. All clusters had positive weights in anterior thalamic, preoptic and hypothalamic nuclei, subiculum, parts of the hippocampus and in the region containing the tuberomammillary nucleus. Clusters 1–3 displayed positive weights in neuromodulatory brainstem regions, substantia nigra, and ventral tegmental area, as well as the entorhinal cortex. The cingulate cortex and retrosplenial cortex and supramammillary nucleus were positive in clusters 2–4. Marked with gray are frames plotted in (**B**). (**B**) Cluster-specific spatial patterns are portrayed on slices selected from **A** (marked with gray rectangles). Characteristic to cluster 1 were positive weights in the dopaminergic substantia nigra and ventral tegmental area as well as in their efferent projections in the nucleus accumbens and caudate-putamen. Positive weighting was also found in the periaqueductal gray and brainstem laterodorsal tegmental and parabrachial nuclei, as well as in the superior colliculus. Cluster 2 was characterized by the strongest positive weights in hypothalamic regions, lateral in particular. Brainstem areas containing the arousal-regulating locus coeruleus, laterodorsal tegmental, and parabrachial nuclei, as well as the septal area and the olfactory tubercle displayed high positive weights. In cluster 3, as in

*Figure 5 continued on next page*

*Figure 5 continued*

cluster 2, the area containing the locus coeruleus, laterodorsal tegmental, and parabrachial nuclei showed positive linkage with pupil dynamics. The highest cluster 3 values were located in preoptic and other hypothalamic areas, as well as in stria terminalis carrying primarily afferent hypothalamic fibers, caudate-putamen, and globus pallidus. In cluster 4, the neuromodulatory region showing the strongest positive weights was the caudal raphe. The anterior parts of the brainstem displayed negative weighting. Characteristic to cluster 4 were high weights in the thalamus and in the hippocampus and the subiculum forming the hippocampal formation. Masked regions (white) did not pass the false discovery rate corrected significance threshold (p=0.01). Abbreviations: Ce – cerebellum, CgCx – cingulate cortex, CP – caudate-putamen, GP – globus pallidus, Hp – hippocampus, Hy – hypothalamus, LC – locus coeruleus, LDT – laterodorsal tegmental nuclei, LH – lateral hypothalamus, NA – nucleus accumbens, OTu – olfactory tubercle, PAG – periaqueductal gray, PB – parabrachial nuclei, PO – preoptic nuclei, Ra – raphe, Sb – subiculum, SC – superior colliculus, Se – septal area, SN – substantia nigra, ST – stria terminalis, Th – thalamus, VTA – ventral tegmental area.

The online version of this article includes the following source data and figure supplement(s) for figure 5:

**Source data 1.** The unmasked cluster maps (A) and the masked maps based on randomization tests with a different random seed (B) are available in the source data file.

**Figure supplement 1.** The spatial map based on cluster 1 trials highlights regions from which pupil-related information was decoded.

**Figure supplement 2.** The spatial map based on cluster 2 trials highlights regions from which pupil-related information was decoded.

**Figure supplement 3.** The spatial map based on cluster 3 trials highlights regions from which pupil-related information was decoded.

**Figure supplement 4.** The spatial map based on cluster 4 trials highlights regions from which pupil-related information was decoded.

lower label match ratios of clusters 1 and 2 in half-split analyses (*Figure 2—figure supplement 4*) should be considered. Our results demonstrate that pupil size changes can be modulated by different combinations of subcortical nuclei, indicating varied brain state fluctuations underlying different oscillatory patterns of pupil dynamics (*Figure 2C*). This is further exemplified by examining the cluster-specific PCA prediction maps. The map of cluster 2 demonstrates the strongest coupling of pupil dynamics with the hypothalamus, which is known to drive pupil dilation (*Ranson and Magoun, 1933*) and also highlights other brain state-regulating nuclei of the ARAS. It is possible that the hypothalamus was the key driver of brain state fluctuation in cluster 2 (*Grimaldi et al., 2014*; *Lee and Dan, 2012*). On the other hand, hypothalamic weights were least prevalent in cluster 1, which displayed strong pupil coupling with the dopaminergic system known to modulate pupil dynamics (*O'Doherty et al., 2003*; *de Gee et al., 2017*; *Shannon et al., 1976*). Finally, in trials of cluster 4, the caudal raphe nucleus was the brainstem neuromodulatory nucleus whose activity had the strongest positive weighting to predict pupil fluctuations. Additionally, the subiculum weights were the strongest in cluster 4 out of all clusters. The positive coupling of the raphe and subiculum hints at the possibility of pupillometry reflecting the activity of circuits responsible for autonomous stress modulation (*Lowry, 2002*). The PCA prediction maps identify key nuclei coupled with pupil dynamics at different states and also highlight the complexity of brain activation patterns responsible for autonomous and brain state regulation.

The presented results should be interpreted in light of employing anesthesia to acquire BOLD fMRI signals within the MRI scanner. Alpha-chloralose was employed due to the quality of BOLD fMRI responses under this anesthetic (*Hyder et al., 2016*; *Alonso et al., 2011*). The neural correlates of brain state-dependent pupil–fMRI correlation differences under alpha-chloralose anesthesia have previously been verified (*Pais-Roldán et al., 2020*). However, as alpha-chloralose has been reported to inhibit the sympathetic system's responses (*Gaumann and Yaksh, 1990*), its influence on pupil size and brain state changes should be investigated similarly to what has been done with other anesthetics. Previous studies report that the use of propofol dampened higher frequency pupil size changes observed in the awake state (*Behrends et al., 2019*), and slow pupil diameter fluctuations were influenced by both isoflurane and urethane anesthesia (*Kum et al., 2016*; *Blasiak et al., 2013*). Furthermore, the brain state changes we observed across trials could be a typical feature of brain activity observed in unanesthetized human subjects e.g. due to arousal or sleep state changes (*Allen et al., 2014*; *Kaufmann et al., 2006*; *Tagliazucchi and Laufs, 2014*) but could also be driven by the anesthetic, as in the case of urethane inducing sleep-like state changes (*Blasiak et al., 2013*; *Clement et al., 2008*). Although the present study is based on the anesthetized rat model, it provides a framework that could be applied to analyze human datasets. Working with awake subjects would mitigate the potential impact of anesthesia on the activity of the sympathetic system, which controls pupillary movements in an antagonistic relationship with the parasympathetic system (*Bonvallet and Zbrozyna, 1963*; *McDougal and Gamlin, 2015*). Additionally, the cognitive component of brain activity reflected

in pupil diameter changes of awake human subjects could be investigated using the PCA-based fMRI decoding method.

Further research should also be directed toward investigating the state-dependent coupling of pupil dynamics and brain activity at finer temporal scales. Importantly, assuming stationarity of the relationship at any scale could lead to oversimplification of the results, as already evidenced by our ability to differentiate four distinct pupil–fMRI coupling patterns instead of one common correlation map. Combining the analysis of individual fMRI frames (*Liu et al., 2013*; *Tseng and Poppenk, 2020*) with the phase of pupil diameter fluctuation, which is known to reflect the activity of different cortical neural populations (*Reimer et al., 2014*), would demonstrate whole-brain activity patterns coupled with pupil dilation and constriction. Finally, regions like the subiculum, which previously have not been linked to pupil dynamics, but displayed strong coupling weights in our study, could guide future electrophysiological studies to reveal novel neuronal regulatory mechanisms underlying pupil dynamics.

## Conclusion

We provided a framework to investigate the brain state-dependent relationship between pupil dynamics and fMRI. The pupil-related brain activity was decoupled from other signal sources based on PCA decomposition and the cluster-specific pupil–fMRI relationship was identified by integrating optimized PCA weighting features using decoding methods. Eventually, distinct subcortical activation patterns were revealed to highlight varied neuromodulatory nuclei corresponding to pupil dynamics.

## Materials and methods

### Animal preparation

All experimental procedures were approved by the Animal Protection Committee of Tübingen (Regierungsprasidium Tübingen; protocol KY12-14) and performed following the guidelines. Pupillometry and fMRI data acquired from 10 male Sprague Dawley rats had been previously published (*Pais-Roldán et al., 2020*). The rats were imaged under alpha-chloralose anesthesia. For details related to the experimental procedures, refer to *Pais-Roldán et al., 2020*.

### fMRI acquisition and preprocessing

All MRI measurements were performed on a 14.1 T/26 cm magnet (Magnex, Oxford) with an Avance III console (Bruker, Ettlingen) using an elliptic trans-receiver surface coil (~2 × 2.7 cm). To acquire functional data, a whole-brain 3D EPI sequence was used. The sequence parameters were as follows: 1 s TR, 12.5 ms TE, 48 × 48 × 32 matrix size, 400 × 400 × 600 µm resolution. Each run had a length of 925 TRs (15 min 25 s). The RARE sequence was used to acquire an anatomical image for each rat. The RARE parameters were as follows: 4 s TR, 9 ms TE, 128 × 128 matrix size, 32 slices, 150 µm in-plane resolution, 600 µm slice thickness, 8 × RARE factor. The data from all rats were spatially co-registered. First, for each EPI run, all volumes were registered to the EPI mean. The EPI means were registered to corresponding anatomical images. To register all data to a common template, all RARE images were registered to a selected RARE image. The obtained registration matrices were then applied to the functional data. A temporal filter (0.002, 0.15 Hz) was applied to the co-registered data. The registration was performed using the AFNI software package (*Cox, 1996*). Principal component analysis (PCA) implemented in the Python scikit-learn library (*Pedregosa, 2011*) was used to reduce the dimensionality of fMRI data for prediction purposes. The PCA time courses were variance normalized before the optimization of linear regression and GRU weights. The functional and anatomical data are available online (*Sobczak et al., 2021b*).

### Pupillometry acquisition and pupil diameter extraction

For each fMRI scan, a video with the following parameters was recorded: 24 bits per pixel, 240 × 352 pixels, 29.97 frames/s, RGB24 format. A customized MRI-compatible camera was used. For details related to the setup, refer to *Pais-Roldán et al., 2020*. The DeepLabCut toolbox (*Mathis et al., 2018*; *Nath, 2018*) was used to extract the pupil position from each video frame. The toolbox's artificial neural network was optimized using 1330 manually labeled images extracted from 74 eye monitoring videos. Training frames were selected using an automated clustering-based DeepLabCut procedure. Four pupil edge points were manually labeled in each training image. Using the trained network, the

four points were located in each recorded frame and their coordinates were used to calculate the pupil diameter as follows:

$$d = \frac{\sqrt{(x_2 - x_1)^2 + (y_2 - y_1)^2} + \sqrt{(x_4 - x_3)^2 + (y_4 - y_3)^2}}{2}$$

Simultaneously, in each video, the eye size was calculated based on manual landmark identification. The eye size was then used to normalize the pupil size, such that pupil diameter values were limited to the <0, 1> range. The pupil diameter signals were averaged over 1 s windows to match the fMRI temporal resolution while reducing noise. Pupillometry time courses were variance normalized before the optimization of linear regression and GRU weights. The time courses are available online (*Sobczak et al., 2021b*).

## Hemodynamic response function convolution

Pupil signals were convolved with HRF kernels with varied peak times to investigate the influence of correcting for the lag between pupil and fMRI signals. The following equation, involving a positive component for the positive response and a negative one for the undershoot, was used to generate the kernels:

$$f(a) = \frac{t^{a-1} e^{-t}}{\Gamma(a)} - 0.3 \frac{t^{11} e^{-t}}{\Gamma(12)},$$

where $a$ is a parameter controlling the peak time, and $\Gamma$ is the gamma function. HRFs with peaks in the <0; 5> s range were used.

## UMAP dimensionality reduction

The uniform manifold approximation and projection (UMAP) (*McInnes et al., 2020*) algorithm was employed to reduce the dimensionality of pupil–fMRI correlation maps before clustering. We used the Python implementation of the algorithm provided by the authors of the method. First, UMAP finds a k-nearest neighbor graph. Based on silhouette scores we set k = 7. To facilitate clustering, we set the minimum allowed distance between points on the low dimensional manifold to 0. We projected the data from the voxel space (n = 20,804) to a 72-dimensional representation, as this was the highest number of dimensions the method permitted given 74 input trials.

## Gaussian mixture model clustering

To cluster the trials in the low dimensional space resulting from the UMAP embedding, we used the expectation-maximization algorithm fitting mixture of Gaussians models to the data (*McLachlan and Basford, 1988*). We used the Python implementation from the scikit-learn library (*Pedregosa, 2011*) with default parameters.

## Silhouette analysis – cluster number verification

To find the number of clusters for successive analyses, we evaluated clustering results using silhouette analysis (*Rousseeuw, 1987*) implemented in the Python scikit-learn library (*Pedregosa, 2011*). For each point, the method computes a silhouette score which evaluates how similar it is to points in its cluster versus points in other clusters. The clustering of the entire dataset was evaluated by computing the mean silhouette score across all points. The clustering result with the highest mean silhouette score was selected for successive analyses.

## Cluster reproducibility

The cluster membership label of each trial was specified based on 100 repetitions of UMAP dimensionality reduction and GMM clustering applied to all trials. We found the final cluster labels by identifying which trials clustered together most often. These final labels were used to create cluster-specific correlation maps. Both the labels and maps were compared with those generated in following analyses to evaluate cluster reproducibility. In particular, we compared label match ratios and cluster map similarities (spatial correlations). We generated alternative clustering results based on: half-split analysis (randomly dividing the 74 trials into groups of 37), using HRF-convolved pupil signals, temporally splitting the data into more trials with shorter durations, and employing spatial surrogates with

properties matching those of real maps. We repeated each analysis 100 times and compared results with the initial clustering.

## Surrogate map generation

Surrogate maps were created using the Brainsmash toolbox (*Burt et al., 2020*). For each correlation map, we generated 100 artificial surrogates that had the same values and spatial autocorrelation as the real map but different spatial patterns. By showing that clustering results based on surrogates were different, we verified that our clustering was not dependent on, e.g., mean map values but on the spatial patterns and regions highlighted in the maps.

## Power spectral density estimation

The spectral analysis was performed using the Python SciPy library (*Virtanen et al., 2020*). To compute the PSDs of utilized signals, we employed Welch's method (*Welch, 1967*), with the following parameters: 512 discrete Fourier transform points; Hann window; 50 % overlap.

## Correlation map-based prediction

Following a strategy described in previous studies (*Pais-Roldán et al., 2020*; *Chang et al., 2016*) we used a pupil–fMRI correlation map to predict pupillometry time courses given fMRI input data. To create the correlation map, pupillometry and fMRI data were concatenated across all trials and the pupil diameter fluctuation signal was correlated with each voxel's signal. This generated a 3D volume (the correlation map), which was then spatially correlated with each individual fMRI volume yielding a single predicted value for each time point. As the resulting time courses' amplitudes were bounded to the $<-1; 1>$ range and not informative of the target signals amplitudes, Pearson's correlation coefficient was used to evaluate the quality of the predictions on a trial-by-trial basis.

## Linear regression variants

Linear regression was used to predict pupillometry data given fMRI-PCA inputs. Four linear regression variants were available to a Bayesian optimizer, which selected both the linear model type and its parameters. The available variants were ordinary least squares, Ridge, Lasso and elastic-net regression models. Python scikit-learn library (*Pedregosa, 2011*) implementations were used. L2 Ridge regression with a regularization parameter $\alpha = 19861$ obtained the best prediction scores and was found using the Hyperopt toolbox (*Bergstra, 2011*; *Bergstra et al., 2013*).

**Table 1.** Optimized GRU hyperparameters.

| Parameter name | Description | Range | Final value |
|---|---|---|---|
| Number of layers | Multiple recurrent layers could be stacked on top of each other. | [1; 3] | 1 |
| Hidden size | Hidden state vector size. | [10; 500] | 300 |
| Learning rate | The rate at which network weights were updated during training. | $[10^{-6}; 1]$ | 0.0023 |
| L2 | Strength of the L2 weight regularization. | [0; 10] | 0.0052 |
| Gradient clipping | Gradient clipping (*Pascanu et al., 2013*) limits the gradient magnitude at a specified maximum value. | [yes; no] | Yes |
| Max. gradient | Value at which the gradients are clipped. | [0.1, 2] | 1 |
| Dropout | During training, a percentage of units could be set to 0 for regularization purposes (*Srivastava et al., 2014*). | [0; 0.2] | 0 |
| Residual connection | Feeding the input directly to the linear decoder bypassing the RNN's computation. | [yes; no] | No |
| Batch size | The number of training trials fed into the network before each weight update. | [3; 20] | 12 |

## GRU

The second model employed for pupillometry decoding was the gated recurrent unit (GRU) (*Cho, 2014*) artificial neural network. The GRU is a recurrent neural network, which encodes each element of the input fMRI-PCA sequence $x$ into a hidden state vector $h(t)$ through the following computations:

$$r(t) = \sigma\left(W_{ir}x(t) + b_{ir} + W_{hr}h(t-1) + b_{hr}\right)$$

$$z(t) = \sigma\left(W_{iz}x(t) + b_{iz} + W_{hz}h(t-1) + b_{hz}\right)$$

$$n(t) = tanh\left(W_{in}x(t) + b_{in} + r(t) \odot \left(W_{hn}h(t-1) + b_{hn}\right)\right)$$

$$h(t) = \left(1 - z(t)\right) \odot n(t) + z(t) \odot h(t-1)$$

where $r, z, n$ are the reset, update, and new gates, $W$ are matrices connecting the inputs, gates, and hidden states, $\sigma()$ and $tanh()$ are the sigmoid and hyperbolic tangent functions, $b$ are bias vectors, and $\odot$ is the elementwise product. A linear decoder generated predictions based on the hidden state vector:

$$y(t) = w_{out}h(t).$$

The correlation coefficient was used as the loss function. The networks were trained in PyTorch (*Paszke, 2019*) using the Adam optimizer (*Kingma and Ba, 2017*). Hyperparameters were found using Bayesian optimization using the tree of Parzen estimators algorithm (Hyperopt toolbox, n = 200) (*Bergstra, 2011*; *Bergstra et al., 2013*). The optimized hyperparameters have been described in *Table 1*. Early stopping was used in the Bayesian optimization procedure. To set the final number of training epochs for the best network, cross-validation was repeated and the GRU was trained for 100 epochs on each split. Training for seven epochs yielded the best performance.

## Cross-validation

The available 74 trials were divided into training (n = 64) and test (n = 10) sets. Linear regression and GRU parameters were found based on the training set with fourfold cross-validation. The final performance was evaluated on the test set. Scores of the correlation-template-based prediction were based on the same data splits.

## Spatial map – linear regression

To create spatial maps highlighting areas that contributed to linear regression predictions, we weighted PCA component maps by their associated linear regression weights, summed them, and took their means. Region borders from the rat brain atlas (*Paxinos and Watson, 2006*) were matched to and overlaid on spatial map slices.

## Spatial map – GRU

To create spatial maps highlighting areas that contributed to GRU predictions, we computed gradients of each of the predicted time points with respect to the 300 input features. We then averaged the gradients across all time points for each of the features and used these mean values just like the weights in the case of linear regression map generation.

## Variance explained

We obtained the fMRI variance explained by each PCA component directly from the scikit-learn (*Pedregosa, 2011*) PCA model. To compute the pupil variance explained by each of the PCA time courses, we used an approach described in *Musall et al., 2019* with fourfold cross-validation. The explained variance of each component was found by randomly shuffling the time points of all other components, training the Ridge linear regression model ($\alpha = 19861$) on shuffled data and assessing the explained variance based on generated predictions.

## Statistical tests – prediction

We used a paired t-test to compare the prediction scores across methods.

## Statistical tests – linear regression spatial maps

To test which linear regression spatial map values significantly contributed to the predictions, we used randomization tests. For each cluster, we shuffled the input and output pairings 10,000 times, trained a linear model, and created a spatial map for each of those pairings. We then compared the values in the original maps with the shuffled ones. Values that were at least as extreme as the shuffled values at the 0.005 positive or negative percentile (p=0.01) were considered significant. The results were controlled for false discovery rate with adjustment (*Benjamini and Hochberg, 1995*; *Yekutieli and Benjamini, 1997*).

## Acknowledgements

We thank Dr. R Pohmann and Dr. K Buckenmaier for technical support; Dr. E Weiler, Dr. P Douay, Mrs. R König, Ms. S Fischer, Ms. H Schulz, and Dr. Jörn Engelmann for animal/lab maintenance and support; and the Analysis of Functional NeuroImages (AFNI) team for software support. Funding: This research was supported by internal funding from Max Planck Society, NIH Brain Initiative funding (RF1NS113278–01, R01MH111438–01) and shared instrument grant (S10 MH124733-01), German Research Foundation (DFG) YU215/2-1 and Yu215/3–1, BMBF 01GQ1702.

## Additional information

### Funding

| Funder | Grant reference number | Author |
|---|---|---|
| Max-Planck-Gesellschaft | internal funding | Xin Yu |
| National Institutes of Health | RF1NS113278-01 | Xin Yu |
| Deutsche Forschungsgemeinschaft | YU215/2-1 | Xin Yu |
| Bundesministerium für Bildung und Forschung | 01GQ1702 | Filip Sobczak Xin Yu |
| National Institutes of Health | R01MH111438-01 | Xin Yu |
| Deutsche Forschungsgemeinschaft | Yu215/3-1 | Xin Yu |
| National Institutes of Health | S10 MH124733-01 | Xin Yu |

The funders had no role in study design, data collection and interpretation, or the decision to submit the work for publication.

### Author contributions

Filip Sobczak, Conceptualization, Data curation, Formal analysis, Investigation, Methodology, Software, Visualization, Writing – original draft, Writing – review and editing; Patricia Pais-Roldán, Kengo Takahashi, Data curation, Investigation, Writing – review and editing; Xin Yu, Conceptualization, Funding acquisition, Project administration, Resources, Supervision, Writing – original draft, Writing – review and editing

### Author ORCIDs

Filip Sobczak  http://orcid.org/0000-0001-9169-0243
Patricia Pais-Roldán  http://orcid.org/0000-0002-9381-3048
Kengo Takahashi  http://orcid.org/0000-0002-3532-1512
Xin Yu  http://orcid.org/0000-0001-9890-5489

### Ethics

All experimental procedures were approved by the Animal Protection Committee of Tübingen (Regierungsprasidium Tübingen; protocol KY12-14) and performed following the guidelines. The rats were imaged under alpha-chloralose anesthesia.

### Decision letter and Author response

Decision letter https://doi.org/10.7554/eLife.68980.sa1
Author response https://doi.org/10.7554/eLife.68980.sa2

## Additional files

### Supplementary files

• Transparent reporting form

### Data availability

All fMRI datasets, as well as the synchronized pupil-size vectors, reported in this paper have been deposited in Zenodo at https://zenodo.org/record/4670277 (https://doi.org/10.5281/zenodo.4670277). Source data for all figures have been uploaded in the system.

The following dataset was generated:

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
