## [Decision Letter]

**Acceptance summary:**

Pupil diameter is used as an index of the brain's arousal system, and has traditionally thought to be a non-invasive index of specific neuromodulatory activity. It is therefore been heavily used as a measure in neuroscience. More recent data suggests a more complex picture whereby a pupil dilation might track cocktail of different neuromodulators. This paper provides firm data supporting this view, and introduces the new view that the make-up of this cocktail changes significantly over time. Pupil dynamics are linked with different neuromodulatory centers over different intervals of time. This is clearly important data across a broad range of human and animal systems neuroscience.

**Decision letter after peer review:**

Thank you for submitting your article "Decoding the brain state-dependent relationship between pupil dynamics and resting state fMRI signal fluctuation" for consideration by *eLife*. Your article has been reviewed by 2 peer reviewers, and the evaluation has been overseen by Timothy Behrens as the Senior and Reviewing Editor. The following individual involved in review of your submission has agreed to reveal their identity: Jan Willem de Gee (Reviewer #2).

After our discussion, we believe the following 3 points from the review are essential. All remaining points are strongly recommended to improve the paper, but are discretionary.

Essential points

From Reviewer 2:

(1) First point in public review – the lag between pupil and BOLD

From Reviewer 3's public review:

(2) assessing reproducibility of the clusters,

(3) potential inter-dependence between train/testing data.

*Reviewer #3 (Recommendations for the authors):*

My specific suggestions for addressing the above points are as follows:

1) Regarding reproducibility and comparison against null models, the authors might refer to the paper Allen et al., 2014 ("Tracking whole-brain connectivity…"), which examined the reproducibility of time-varying fMRI connectivity matrices using a split-halves analysis, and compared the results against "null" data that matched the actual data in terms of mean, variance, and autocorrelation.

2) In the power spectral density (PSD) of pupil dynamics for each cluster (Figure 2c), it may be helpful to display some measure of variability over the trials within each cluster -- otherwise it is unclear whether the spectral profiles are distinct across the clusters. Also, how should these different spectral peaks be interpreted in light of the corresponding fMRI spatial maps?

3) Regarding the decoding models – from what I understood, the PCA step included all 74 scans, including the test data, which would be a source of leakage between train/test sets. It was also not clear whether trials from the same rat could appear in both training and testing sets, which would also affect the interpretation of the decoding accuracy.

4) It is also mentioned that models were trained on 64 trials, and tested on 10 held-out trials. Since the result might be quite sensitive to the 10 particular trials that were held out, it may be helpful to do a nested cross-validation, holding out multiple subsets of trials for testing (and averaging over the resulting test-set errors).

5) On first reading, I was a bit confused that the paper first shows 1) that different pupil-fMRI correlation patterns were found in different scans, and then 2) that generalizable models (including linear models) could be trained to predict pupil waveforms from fMRI. The section beginning on p. 9 seems to link these sections by fitting models separately across clusters, but it may be helpful to make this conceptually clearer at the outset.

6) As clustering is currently performed over 15-min intervals (entire trials), I wondered if the authors had experimented with using shorter intervals. In other words, based on the mechanisms that could drive these changes in pupil-fMRI correlation (and accounting for limitations of fMRI temporal resolution), how quickly would the correlation patterns expect to shift?

---

## [Author Response]

Essential pointsFrom Reviewer 2:(1) First point in public review – the lag between pupil and BOLDFrom reviewer 3's public review:(2) assessing reproducibility of the clusters,(3) potential inter-dependence between train/testing data.

We thank the reviewers for the positive feedback and for providing ways of improving the study. We addressed the issues pointed out by the reviewers to the best of our capability, in particular, the three essential points as highlighted.

Reviewer #3 (Recommendations for the authors):My specific suggestions for addressing the above points are as follows:1) Regarding reproducibility and comparison against null models, the authors might refer to the paper Allen et al., 2014 ("Tracking whole-brain connectivity…"), which examined the reproducibility of time-varying fMRI connectivity matrices using a split-halves analysis, and compared the results against "null" data that matched the actual data in terms of mean, variance, and autocorrelation.

The point has been addressed in the public review response, however, here we provide an explanation for using the spatial surrogates instead of the temporal ones employed in Allen et al., 2014. We perform dimensionality reduction and clustering on spatial correlation maps. Using temporal surrogates from Allen et al., (2014) did not provide an adequate null model, as (a) applying the same phase shifts to both pupil and fMRI data resulted in creating almost the same spatial correlation maps and the same final results and (b) applying random phase shifts to pupil and fMRI signals before correlating them resulted in close to 0 correlation values and completely random maps. On the other hand, using the Brainsmash toolbox (Burt et al., (2020)), we generated surrogate maps with preserved spatial autocorrelation and value distribution, which allowed us to show that it’s not those two features but the region-specific spatial patterns that drive the clustering.

2) In the power spectral density (PSD) of pupil dynamics for each cluster (Figure 2c), it may be helpful to display some measure of variability over the trials within each cluster -- otherwise it is unclear whether the spectral profiles are distinct across the clusters. Also, how should these different spectral peaks be interpreted in light of the corresponding fMRI spatial maps?

We now plot the standard deviation of the cluster-specific PSD values (Figure 2C). We also included plots of all individual PSDs and example raw pupil signal data (Figure 2 —figure supplement 2). As the different spectral peaks originated from the clustering analysis, they could indicate varied neuromodulatory mechanisms, which were presented in the corresponding fMRI spatial maps. One of our current work is aiming to elucidate the neuronal basis of the varied slow oscillatory features of pupil dynamics based on electrophysiological recordings.

3) Regarding the decoding models – from what I understood, the PCA step included all 74 scans, including the test data, which would be a source of leakage between train/test sets. It was also not clear whether trials from the same rat could appear in both training and testing sets, which would also affect the interpretation of the decoding accuracy.

Point addressed in the public review response.

4) It is also mentioned that models were trained on 64 trials, and tested on 10 held-out trials. Since the result might be quite sensitive to the 10 particular trials that were held out, it may be helpful to do a nested cross-validation, holding out multiple subsets of trials for testing (and averaging over the resulting test-set errors).

Point addressed in the public review response.

5) On first reading, I was a bit confused that the paper first shows 1) that different pupil-fMRI correlation patterns were found in different scans, and then 2) that generalizable models (including linear models) could be trained to predict pupil waveforms from fMRI. The section beginning on p. 9 seems to link these sections by fitting models separately across clusters, but it may be helpful to make this conceptually clearer at the outset.

We now clarified the connection between the decoding analysis and the clustering results at the beginning of the decoding analysis section (Page 5).

6) As clustering is currently performed over 15-min intervals (entire trials), I wondered if the authors had experimented with using shorter intervals. In other words, based on the mechanisms that could drive these changes in pupil-fMRI correlation (and accounting for limitations of fMRI temporal resolution), how quickly would the correlation patterns expect to shift?

Based on the comment, we split the data into shorter trials and repeated the clustering analysis. Based on the silhouette score criterion, selecting n=4 clusters holds up to the length of ~300 s (Figure 2 —figure supplement 6).